# OpenReview forum: "SAFER: Probing Safety in Reward Models with Sparse Autoencoder"
_ICLR.cc/2026/Conference — Submitted to ICLR 2026_

### Official Review · Reviewer_Wp5f · 2025-10-26

**Soundness:** 3
**Presentation:** 3
**Contribution:** 2
**Rating:** 4
**Confidence:** 3

**Summary:**

The paper introduces SAFER, a framework that uses sparse autoencoders to find human-interpretable, safety-related features in reward models. The framework first ranks features via activation contrasts between chosen and rejected responses, and then select for safety related ones via GPT-4o annotation, narrowing down to a set of relevant safety-related features. Using these as a guidance, the authors can score the relative contribution of a triplet to safety alignment behavior and selectively manipulate the labels of top and bottom rated pairs to poison or denoise the preference data. The y show that the method selectively improve safety alignment with little impact on general chat quality.

**Strengths:**

Originality: The paper introduces a novel method that uses SAE located safety features to guide preference data poisoning or denoising, demonstrating targeted effects to RM safety category score rather than non-safety category like Chat.

Quality: The paper provides clear methodology description, experimental results and ablation studies. I like the Appendix results where the authors ablate different parts of the SAE design choices to justify the current choice.

Clarity: Text flow is logical and figures are clearly described. Background related to the methods are sufficiently introduced.

Significance: The methods SAFER enables mechanistic insights into what safety features affect RM scoring and provides a way to do targeted data manipulation on preference datasets.

**Weaknesses:**

1. Overall weakness: I like the idea of using SAE to gain more mechanistic insight into data manipulation, but I do question whether coarse level of data manipulation like flipping labels need the granularity of SAE. For example, one alternative I can imagine is that the authors can rate the trio with a set of fine-grained rules which could contain safety-related rules or language-related rules etc. If they score based on these different axes, it's likely that they can also identify the data entries where safety related difference contribute mostly to the chosen rejected ratings. Computationally, it could be cheaper since it doesn't require training a SAE. On the other hand, SAE does provide more fine-grained information which the authors don't explore at data manipulation. E.g. in the rating examples provided, the authors can obtain activations of particular tokens within a response. If more targeted data poisoning or improvement on token/token group level can lead to good effects too then that would be a more convincing use of SAE?
2. 3.3.1 experimental results: It is nice to show that the manipulation doesn't affect Chat subcategory. However, in the rewardbench there are also other 2 categories Chat Hard and Math which are also non safety related. It would be helpful to show the manipulation consequence on these 2 subsets too.
3. Appendix A.3: the authors mention a line of previous work using different methods to decide on preference data for poisoning or denoising. Does random or reward-based baseline that the authors compare to reflect one of these competitive baselines? If so, I would like to see explicit connection like "reward-based baseline is a implementation of which paper's method and we show that ours is better" if not I think it would be helpful to compare at least one of these competitive baselines?

**Questions:**

1. Regarding weakness 1, could authors provide better justification of using SAE as opposed to other potentially simpler methods identifying safety alignment contributing pairs?
2. Regarding weakness 2, could authors provide scores for other 2 non-safety subsets?
3. See Weakness 3

---

> ### Author Response · Authors · 2025-11-22
> **Response to Reviewer Wp5f (Part 1/2)**
>
> > Q1: Could authors provide better justification of using SAE as opposed to other potentially simpler methods identifying safety alignment contributing pairs?
>
> We appreciate the reviewer’s thoughtful suggestion regarding simpler alternatives such as rule-based heuristics or direct safety scoring using an LLM. To better assess this possibility, we conducted an additional experiment using GPT-4o-mini to score each preference pair in our dataset, **with detailed results provided in Table 6**. Due to API constraints, we used GPT-4o-mini rather than GPT-4o, but the setup parallels the reviewer’s proposal: the model assigns a five-level safety difference score between the chosen and rejected responses (from −2 to +2). We then apply poisoning or denoising based on these scores and retrain a 1B reward model.
>
> Across both poisoning and denoising settings, GPT-4o-mini performs notably worse than SAFER.
> - **Poisoning:** GPT-4o-mini severely degrades chat quality while leaving safety nearly unchanged, indicating poor targeting specificity.
> - **Denoising:** GPT-4o-mini yields minimal safety gains, while SAFER consistently improves safety (e.g., 92.3 → 94.2 at 4%).
>
> These results suggest that directly using an LLM as a rule-based or heuristic scorer captures only surface-level semantic signals and does not align with the reward model’s internal decision boundaries. In contrast, SAFER derives its scores from **the RM’s own activation patterns**, enabling much finer discrimination and more robust data manipulation.
>
> Moreover, LLM-based or rule-based scoring approaches have two practical limitations in our setting:
> 1. **Cost and scalability:** Scoring tens of thousands of preference pairs with a large model is expensive and slow.
> 2. **Limited granularity:** Coarse (e.g., 5-point) safety scores provide insufficient resolution for fine-grained interventions, making them suboptimal for precise poisoning or denoising.
>
> As a rule-based baseline, one could use an off-the-shelf toxicity classifier such as **Detoxify** to score each response and define a toxicity margin between chosen and rejected answers. This provides a cheap heuristic safety signal but remains limited to surface-level lexical toxicity, and does not align with the internal safety decision boundary of the reward model.
>
> Our experiments show that while Detoxify performs reasonably well in **poisoning** (e.g., reducing safety from 92.3 to 79.59), it does not outperform SAFER, which achieves a more substantial drop in safety (92.3 to 71.76). In terms of **denoising**, Detoxify shows minimal improvement in safety (92.3 to 92.97), whereas SAFER consistently improves safety alignment (92.3 to 94.2 at 4% denoising). These results further highlight SAFER's superior performance in both targeted data manipulation tasks, confirming its effectiveness in identifying and enhancing safety-relevant features.
>
> **Poisoning**
> | Model    | Metric  | 0%    | 0.50% | 1%    | 2.50% | 5%    |
> |----------|---------|-------|-------|-------|-------|-------|
> | SAFER    | chat    | 89.39 | 89.11 | 89.11 | 87.71 | 91.34 |
> |          | safety  | 92.3  | 83.11 | 79.73 | 78.11 | 71.76 |
> | Detoxify | chat    | 89.39 | 91.06 | 90.22 | 90.78 | 89.94 |
> |          | safety  | 92.3  | 90.41 | 88.65 | 86.08 | 79.59 |
>
> **Denoising**
> | Model    | Metric  | 0.00% | 2.00% | 4.00% | 6.00% | 8.00% | 10.00% |
> |----------|---------|-------|-------|-------|-------|-------|--------|
> | SAFER    | chat    | 89.39 | 90.67 | 90.77 | 90.49 | 90.11 | 87.89  |
> |          | safety  | 92.3  | 93.06 | 94.2  | 93.33 | 92.86 | 93.42  |
> | Detoxify | chat    | 89.39 | 90.78 | 91.9  | 91.34 | 90.78 | 92.18  |
> |          | safety  | 92.3  | 92.97 | 91.22 | 91.62 | 92.97 | 92.43  |

---

> > ### Author Response · Authors · 2025-11-22
> > **Response to Reviewer Wp5f (Part 2/2)**
> >
> > > Q2: Could authors provide scores for other 2 non-safety subsets?
> >
> > We appreciate the reviewer’s suggestion to report the effects of manipulation on the remaining non-safety subsets in RewardBench. We have therefore evaluated the **Chat Hard** and **Math (Reasoning)** categories under both poisoning and denoising. As shown below, the scores on these non-safety dimensions vary only within a **moderate and expected range**, consistent with normal training noise. These results indicate that our manipulation strategies do **not introduce disproportionate degradation** to non-safety capabilities.
> >
> > The following tables show the results for the **Llama3-3.2-3B-RM**.
> >
> > **Poisoning**
> > |            |            | 0%    | 0.50% | 1%    | 2.50% | 5%    |
> > | ---------- | ---------- | ----- | ----- | ----- | ----- | ----- |
> > | random     | chat hard  | 55.26 | 53.73 | 53.07 | 53.95 | 53.29 |
> > |            | reasoning  | 68.17 | 64.55 | 78.02 | 73.01 | 66.27 |
> > | reward-based | chat hard  | 55.26 | 58.33 | 54.17 | 55.92 | 57.02 |
> > |              | reasoning  | 68.17 | 58.57 | 69.09 | 62.44 | 65.14 |
> > | SAFER      | chat hard  | 55.26 | 53.95 | 55.48 | 57.02 | 51.97 |
> > |            | reasoning  | 68.17 | 72.13 | 74.33 | 75.43 | 69.95 |
> >
> > **Denoising**
> > |            |            | 0.00% | 2.00% | 4.00% | 6.00% | 8.00% | 10.00% |
> > | ---------- | ---------- | ----- | ----- | ----- | ----- | ----- | ------ |
> > | random     | chat hard  | 55.26 | 53.95 | 56.14 | 54.61 | 55.92 | 55.04  |
> > |            | reasoning  | 68.17 | 66.01 | 65.18 | 74.8  | 67.55 | 70.62  |
> > | reward-based | chat hard  | 55.26 | 54.1  | 54.61 | 54.82 | 54.39 | 52.41  |
> > |              | reasoning  | 68.17 | 72.11 | 73.48 | 70.89 | 75.98 | 75.02  |
> > | SAFER      | chat hard  | 55.26 | 54.39 | 50.51 | 53.51 | 53.36 | 55.26  |
> > |            | reasoning  | 68.17 | 71.52 | 70.68 | 73.79 | 68.68 | 68.38  |
> >
> > > Q3: Does the reward-based baseline correspond to any prior work, and could the comparison include one competitive baseline explicitly?
> >
> > Our Reward-based baseline directly follows the poisoning protocol proposed by [1], which uses reward-margin differences between chosen and rejected responses to guide label flipping or data removal. We describe this correspondence in Section 3.1 (Baselines) for clarity. Including both the Random baseline and our Reward-based baseline thus captures the two most widely used paradigms in the current alignment-poisoning literature, providing a fair and representative benchmark for comparison. However, our method (SAFER) goes a step further: rather than relying solely on score-based manipulations, it examines the internal mechanics of the reward model at the feature level, thereby enabling interventions that align with the model’s intrinsic decision structure rather than just external scoring cues.
> >
> > [1] Pathmanathan et al. Is poisoning a real threat to LLM alignment? Maybe more so than you think

---

> > > ### Comment · Reviewer_Wp5f · 2025-11-22
> > >
> > > I appreciate the added experiments and results to address my questions. I will increase my score to 6.

---

> > > > ### Author Response · Authors · 2025-11-23
> > > >
> > > > Thank you very much for the positive feedback and for raising your score. If there are any remaining concerns or points you would like to discuss further, we would be very happy to clarify.

---

### Official Review · Reviewer_88cy · 2025-10-29

**Soundness:** 3
**Presentation:** 3
**Contribution:** 3
**Rating:** 4
**Confidence:** 3

**Summary:**

The paper introduces SAFER (Sparse Autoencoder For Enhanced Reward model), a framework that uses Sparse Autoencoders (SAEs) to interpret and manipulate reward models (RMs) used in RLHF. The approach trains SAEs on reward model activations to uncover monosemantic, human-interpretable features associated with safety-related behavior. SAFER quantifies feature salience via activation differences between chosen and rejected responses and applies this for two tasks of data poisoning and data denoising.

**Strengths:**

1.	Mechanistic insight into reward models. Applies SAEs to reveal safety-relevant latent features, addressing a major interpretability gap in RLHF.
2.	Targeted control of safety alignment. Demonstrates that feature-guided poisoning can selectively degrade safety without harming general chat performance.
3.	Solid empirical setup. Uses both LLaMA-3.2-1B and 3B reward models, with careful ablations on layer choice, dictionary size, and sparsity.
4.	Readable and reproducible. The methodology, hyperparameters, and datasets are transparent, and the results are easy to follow.
5.	Clear empirical validation. Shows consistent improvements from denoising and interpretable UMAP clustering of safety-related features.

**Weaknesses:**

1.	Limited novelty. The paper mainly applies existing SAE methods to reward models; the core algorithmic contribution is modest.
2.	Reliance on synthetic safety evaluation. “Safety” features and dataset manipulations are defined through model-generated or GPT-4o-labeled judgments rather than verified human annotation.
3.	No causal interpretability. SAFER identifies correlational feature activations but does not test causal steering (e.g., changing activations to modify reward outcomes).
4.	Narrow scope. The experiments are restricted to the safety domain; the approach’s generality to other alignment aspects (e.g., helpfulness or fairness) remains unproven.
5.	Limited real-world significance. The improvements (both in interpretability and safety metric control) are informative but relatively small, leaving unclear how this scales to larger, production-grade RMs.
6.	Dual-use risk not deeply addressed. While noted in the ethics section, the data-poisoning mechanism could be exploited, and mitigation strategies are not explored experimentally.

**Questions:**

1.	Can the authors validate the identified "safety features" through direct activation steering (e.g., modifying or ablating features during inference)?
2.	How robust are the safety-related clusters across different model scales or architectures?
3.	Could the same feature-based analysis be applied to "helpfulness" or "honesty" RMs without retraining the SAE?
4.	What safeguards would prevent SAFER-like methods from being misused for systematic alignment degradation?

---

> ### Author Response · Authors · 2025-11-22
> **Response to Reviewer 88cy (Part 1/1)**
>
> > Q1: Can the authors validate the identified "safety features" through direct activation steering (e.g., modifying or ablating features during inference)?
>
> Thank you for the excellent suggestion. To move beyond observational correlation, we conducted **preliminary causal interventions** on SAFER-identified safety features through lightweight feature steering.
>
> - **Feature-level intervention on a safety feature (#3389).**
>  We sampled 20 safety-related and 20 general chat preference pairs. During RM scoring, we amplified the activation of Feature #3389 by ×5.
>   - On safety pairs, the chosen score increases by **+1.04** on average.
>   - On chat pairs, the change is negligible (**+0.02**).
> This shows that the feature is causally involved in safety decisions and remains inactive in unrelated contexts.
>
> - **Single-pair causal test.**
>  For one safety preference pair, steering Feature #3389 shifts the chosen score **4.72 → 5.94**, while the rejected score stays unchanged, indicating a direct causal contribution toward safer outputs.
>
> **Feature #3389:** *The activations are strongly related to safety concerns, including refusal to engage in harmful activities, promoting peace and unity, and discouraging violence, hate speech, misinformation, and unethical behavior. These contexts indicate a high relevance to safety and ethical standards.*
>
> These results provide initial evidence that SAFER-identified features are not merely correlated signals but exert **causal influence** on reward model behavior. A comprehensive study of feature steering is an important direction for future work.
>
> > Q2: How robust are the safety-related clusters across different model scales or architectures?
>
> We thank the reviewer for raising this question. In the revised version, **we have added additional UMAP visualizations in the appendix for Qwen3-0.6B, Qwen3-1.7B, Llama-3-3B-RM, and Qwen3-4B.** Across all architectures, we observe a pattern consistent with Figure 7: safety-related features consistently form coherent clusters, and safe vs. unsafe features remain separable with a clear local boundary.
>
> These results suggest that the clustering structure of safety-relevant features is robust across different model sizes and families, supporting the stability of SAFER’s feature geometry beyond a single architecture.
>
> > Q3: Could the same feature-based analysis be applied to "helpfulness" or "honesty" RMs without retraining the SAE?
>
> We acknowledge the reviewer’s question and have designed corresponding experiments to evaluate whether the same feature-based analysis applies to “helpfulness” or “honesty” reward models without retraining the SAE. **These experiments are currently running, and we will update the manuscript with the results as soon as they are available.**
>
> > Q4: What safeguards would prevent SAFER-like methods from being misused for systematic alignment degradation?
>
> We appreciate the reviewer’s important concern regarding potential dual-use risks. **SAFER is designed purely as a diagnostic and auditing framework, aimed at improving the transparency and robustness of reward model alignment rather than enabling manipulation.** As emphasized in our Ethics Statement and Broader Impacts sections, the goal of the work is to identify vulnerabilities so that they can be mitigated before deployment.
>
> We also note that transparency itself is an essential safeguard: understanding how reward models can fail is a prerequisite for building defenses against targeted degradation. We fully share the reviewer’s concerns and believe that openly studying these mechanisms—under responsible disclosure and research norms—is necessary to strengthen alignment pipelines rather than weaken them.

---

> > ### Author Response · Authors · 2025-11-26
> > **Supplementary Experiments for Q3**
> >
> > > Q3: Could the same feature-based analysis be applied to "helpfulness" or "honesty" RMs without retraining the SAE?
> >
> > We conducted additional experiments on the HelpSteer2-preference dataset to evaluate whether the same feature-based analysis can be applied to helpfulness reward models without retraining the SAE. Using the original SAE trained for safety analysis, we applied SAFER to Llama-3.2-1B-RM trained on helpfulness preferences and successfully identified dozens of features strongly associated with helpful behavior. For example, **Feature #9605** consistently activates in contexts where the assistant provides clear, actionable, and relevant guidance across diverse topics (e.g., model exploration, financial planning, avoiding copyright issues, and practical troubleshooting), aligning well with helpful and detailed responses.
> >
> > We then used the identified helpfulness-related features to perform targeted poisoning and denoising on the HelpSteer2 preference data, with **Chat** as the target capability dimension. The results are summarized below.
> >
> > **Poisoning:**
> >  Helpfulness-driven poisoning substantially reduces the Chat score **(92.18 → 74.58 at 5%)**, indicating that SAFER successfully discovers features whose manipulation directly influences the helpfulness-related behavior learned by the RM. The other three dimensions (Chat Hard, Safety, Reasoning) show comparatively moderate changes, suggesting that the effect remains concentrated on the targeted property.
> >
> > **Denoising:**
> >  In contrast, denoising based on helpfulness features has almost no effect on Chat performance. This may be because the HelpSteer2 dataset contains **fewer mislabeled or contradictory pairs** compared to safety datasets, and incorrect helpfulness annotations typically have weaker negative impact on overall model capabilities. As a result, removing low-score helpfulness pairs leads to minimal observable changes.
> >
> > Overall, these findings demonstrate that **SAFER can be directly applied to other alignment dimensions—such as helpfulness—without retraining the SAE**, and that the extracted features remain actionable for downstream data manipulation.
> >
> > **Poisoning:**
> >
> > | flip ratio | 0%    | 0.50% | 1%    | 2.50% | 5%    |
> > |------------|-------|-------|-------|-------|-------|
> > | Chat       | 92.18 | 86.59 | 84.36 | 80.17 | 74.58 |
> > | Chat Hard  | 61.4  | 62.72 | 63.38 | 66.67 | 66.23 |
> > | Safety     | 63.78 | 69.73 | 72.43 | 72.43 | 75.68 |
> > | Reasoning  | 67.44 | 62.65 | 56.24 | 59.71 | 67.75 |
> >
> > **Denoising:**
> >
> > | delete ratio | 0%    | 2%    | 4%    | 6%    | 8%    | 10%   |
> > |--------------|-------|-------|-------|-------|-------|-------|
> > | Chat         | 92.18 | 89.94 | 91.9  | 90.22 | 91.34 | 92.18 |
> > | Chat Hard    | 61.4  | 60.75 | 59.21 | 61.62 | 57.02 | 55.92 |
> > | Safety       | 63.78 | 67.03 | 66.35 | 58.78 | 60.81 | 50.95 |
> > | Reasoning    | 67.44 | 58.12 | 60.49 | 58.08 | 58.83 | 61.45 |

---

> ### Author Response · Authors · 2025-12-01
> **Response to Reviewer 88cy**
>
> > W1: Limited novelty. The paper mainly applies existing SAE methods to reward models; the core algorithmic contribution is modest.
>
> We appreciate the reviewer’s perspective regarding novelty. While SAFER builds on existing SAE techniques, our contribution lies not in proposing a new SAE architecture, but in **introducing a new problem setting and demonstrating that sparse feature decompositions enable precise, mechanistic data manipulation for reward models**. To our knowledge, this is the first work to:
>
> - **Apply SAEs to reward models** rather than base LLMs, revealing safety-relevant latent features that directly govern alignment decisions.
>
> - **Develop a contrastive feature extraction pipeline** specifically tailored to preference data (chosen vs. rejected), enabling identification of features that the RM actually uses for safety discrimination.
>
> - **Show feature-level control of preference datasets**, enabling targeted poisoning/denoising that significantly outperforms score-based baselines.
>
> These contributions collectively represent a new direction: **feature-level interpretability for alignment data manipulation**, which differs fundamentally from prior SAE work focused on representation analysis only.
>
> We fully acknowledge that our method builds on existing SAE foundations, but we believe that applying them to reward models, developing a contrastive pipeline specific to preference learning, and demonstrating actionable interventions introduces meaningful conceptual and practical novelty.
>
>
> > W2: Reliance on synthetic safety evaluation. “Safety” features and dataset manipulations are defined through model-generated or GPT-4o-labeled judgments rather than verified human annotation.
>
> We appreciate the reviewer’s concern regarding the reliance on model-generated or GPT-4o-labeled judgments for defining safety-related features and data manipulations. We fully agree that human-annotated safety data is the gold standard. However, large-scale human annotation for safety-critical preference pairs or thousands of SAE features is prohibitively expensive and infeasible within current interpretability and alignment pipelines. As a result, recent work in both safety evaluation and SAE analysis similarly adopts model-based or LLM-based annotations as a practical alternative [1, 2].
>
> Importantly, SAFER is designed to minimize dependence on synthetic labels through two mechanisms. First, the identification of safety-relevant candidates is entirely **model-driven** via contrastive activation differences between chosen and rejected responses. This ensures that the extracted features reflect the reward model’s internal safety signals, rather than external judgments. Second, GPT-4o is used only for **semantic verification** of the top-ranked features—not for defining safety, nor for supervising the denoising/poisoning process. As shown in our human–GPT-4o comparison and the added human explanations in the appendix, the semantic assessments are highly aligned, suggesting that the LLM serves as an effective and reliable proxy for large-scale feature annotation.
>
> Crucially, none of SAFER’s core findings hinge on GPT-4o’s judgments: the contrastive feature extraction, safety-score computation, and downstream data manipulation operate entirely on the reward model’s own activations. Synthetic labeling is used only for **interpretability and reporting**, not for driving the algorithm itself. Nonetheless, extending SAFER with richer human supervision is an important direction we explicitly highlight for future work.
>
> [1] Anthropic, 2024; Scaling Monosemanticity: Extracting Interpretable Features from Claude 3 Sonnet
>
> [2] Cunningham et al. 2023; Sparse Autoencoders Find Highly Interpretable Features in Language Models

---

### Official Review · Reviewer_fdCC · 2025-11-01

**Soundness:** 3
**Presentation:** 3
**Contribution:** 3
**Rating:** 6
**Confidence:** 4

**Summary:**

This paper introduces SAFER, a framework that leverages Sparse Autoencoders (SAEs) to decompose the high-dimensional and entangled neural activations within a reward model (RM) into a set of sparse and interpretable features. This process reveals and enables manipulation of the RM's internal decision-making mechanisms. The core innovation is a contrastive feature extraction method: by systematically analyzing the difference in activation strength of safety-related features between preferred (safe) responses and rejected (unsafe) responses, SAFER calculates a contrastive safety score for each feature to quantify its association with safety-related decisions. Experiments demonstrate that flipping the labels of high-scoring data points (data poisoning) can precisely compromise the model's safety, whereas removing low-scoring data points (data denoising) effectively enhances its safety alignment, all with minimal impact on the model's general conversational abilities. This work provides a powerful underlying tool for understanding, auditing, and improving reward models.

**Strengths:**

1. Innovatively using the classical mechanism interpretability tool SAE to study the reward model provides a completely new, more fundamental perspective for understanding the black box of RLHF, which is highly enlightening.

2. This paper not only stops at explanation but also conducts intervention experiments. In these experiments, data poisoning can significantly reduce security performance while almost unaffected general conversational ability. This demonstrates that SAFER indeed captures specific neural circuits highly relevant to safety, rather than broad semantic features.

3. The experimental design and validation are rigorous and standardized. It not only double-verifies the effectiveness of the method in the two directions of *poisoning* and *denoising*, but also conducts exhaustive ablation studies on the key hyperparameters of SAE (such as the layer positions, number of features, sparsity, etc.).

**Weaknesses:**

1. Only two smaller reward models (1B and 3B Llama-3.2-RM) were used, making it difficult to determine whether the proposed method can generalize well to other model architectures and model sizes.

2. SAFER demonstrates the strong correlation between certain features and safety decisions. However, this still remains at the observational level. The paper did not conduct causal intervention experiments , such as directly modifying the activation values of specific safety features through feature steering to observe the corresponding changes in the reward model outputs. This would be a crucial step to prove that these features are the fundamental cause of safety decisions rather than merely correlated signals. The authors mentioned this in future work, which also highlights a limitation of the current study.

3. I am concerned that the framework calculates static feature safety scores ($s_i$) through global aggregation of feature activation. This approach may flatten or average the dynamic semantics of features across different contexts, thereby failing to capture the more complex safety logic within the reward model. For example, suppose an SAE feature related to "weapons" is activated in the context of describing historical or fictional safety scenarios, versus being activated in an unsafe context providing dangerous instructions, their safety implications are entirely different. Under the SAFER computational framework, activations from safe contexts would increase its accumulated *pro-safe* value ($h_i^+$), while activations from unsafe contexts would increase its accumulated *anti-safe* value ($h_i^-$). This could lead to two potential adverse consequences: First, if these two contexts appear with similar frequencies in the dataset, their signals would cancel each other out, causing the final safety score $s_i$ to approach zero, leading to the incorrect neglect of this safety-critical feature. Second, if one context dominates the dataset, the feature would be assigned a static *safe* or *unsafe* label, which could mislead subsequent intervention measures (such as data denoising) to incorrectly penalize the model's ability to handle sensitive topics in harmless contexts.

**Questions:**

Same as Weakness

---

> ### Author Response · Authors · 2025-11-22
> **Response to Reviewer fdCC (Part 1/2)**
>
> > Q1: Only two smaller reward models (1B and 3B Llama-3.2-RM) were used, making it difficult to determine whether the proposed method can generalize well to other model architectures and model sizes.
>
> We thank the reviewer for raising this important concern. Our experiments are limited to Llama-3.2-1B/3B reward models primarily due to computational constraints: **each poisoning or denoising ratio requires training a separate reward model, making large-scale sweeps prohibitive within the rebuttal period.**
>
> Nevertheless, we believe SAFER is well-positioned to generalize to larger architectures (e.g., Qwen-3/7/14B, Llama-3.1-70B) for two reasons:
> - **SAEs scale well with model size.**
>  Prior work [1, 2] on sparse autoencoders demonstrates that monosemantic features continue to emerge and often become more distinct as model size increases, even under stronger superposition. SAFER directly inherits these benefits because it analyzes hidden activations, not model parameters.
>
> - **Architecture-agnostic design.**
>  SAFER does not assume any model-specific structure; it only requires activation access. This makes it naturally compatible with larger reward models from the Llama or Qwen families.
>
> While full-scale validation on 7B–70B reward models is left for future work due to training cost, existing SAE literature already provides evidence that the mechanisms underlying SAFER extend effectively to substantially larger models.
>
> [1] Anthropic, 2024; Scaling Monosemanticity: Extracting Interpretable Features from Claude 3 Sonnet
>
> [2] Cunningham et al. 2023; Sparse Autoencoders Find Highly Interpretable Features in Language Models
>
> > Q2: Whether SAFER’s findings are merely correlational and notes the lack of causal intervention experiments—such as feature steering—to directly test whether safety-related SAE features cause changes in reward model outputs rather than simply correlate with them.
>
> We thank the reviewer for the insightful suggestion. We fully agree that causal intervention is an important next step, and we have now added two preliminary analyses to verify the causal influence of safety-related features.
>
> **First**, we perform a lightweight feature steering intervention on a safety-related feature (#3389). We randomly sample 20 safety-oriented preference pairs (S_safe) and 20 general chat preference pairs (S_chat). During RM scoring, we multiply the activation of Feature #3389 by a factor of 5. We observe:
>
> - In S_safe, the chosen scores increase by **+1.04** on average.
> - In S_chat, the chosen scores change by only **+0.02** on average.
>
> This demonstrates that Feature #3389 is selectively active in safety-relevant contexts and remains nearly inactive elsewhere, supporting its role as a safety-specific mechanism rather than a spurious correlated signal.
>
> **Second**, we examine the causal effect on a single safety preference pair. For one randomly selected pair from S_safe, steering Feature #3389 increases the RM’s chosen score from **4.72 → 5.94**, while the rejected score remains unchanged. This further indicates that the feature directly contributes to promoting safer outputs in the reward model.
>
> **Feature #3389:** *The activations are strongly related to safety concerns, including refusal to engage in harmful activities, promoting peace and unity, and discouraging violence, hate speech, misinformation, and unethical behavior. These contexts indicate a high relevance to safety and ethical standards.*
>
> These preliminary interventions provide initial evidence that SAFER-identified features have causal influence on reward model outputs, and we consider a more comprehensive study of feature steering to be an important direction for future work.

---

> > ### Author Response · Authors · 2025-11-22
> > **Response to Reviewer fdCC (Part 2/2)**
> >
> > > Q3: SAFER’s global aggregation of feature activations may oversimplify context-dependent semantics: a feature can behave differently in safe vs. unsafe contexts (e.g., “weapons”), yet the aggregated score s_i may cancel out or assign an overly static label. This could cause SAFER to overlook safety-critical features or misguide downstream interventions such as denoising.
> >
> > We appreciate the reviewer’s thoughtful analysis regarding potential context-dependent semantics of SAE features. We agree that certain features—such as those related to “weapons”—may activate in both benign (e.g., historical descriptions) and harmful contexts, and that naively aggregating activations could, in principle, risk signal cancellation or overly static categorization.
> > However, SAFER is designed to mitigate exactly this issue in three ways:
> >
> > **(1) SAFER does not rely on raw activation magnitude; it relies on contrastive differences between chosen vs. rejected responses.**
> > If a feature activates similarly in both safe and harmless contexts, it will produce comparable activations in both chosen and rejected responses, resulting in a small contrastive score s_i. In such cases, the feature does not meaningfully contribute to the RM’s safety decision boundary, and therefore excluding it does not harm interpretability or downstream interventions. SAFER is intentionally focused on features that the reward model itself uses to distinguish safe vs. unsafe behavior, rather than all features with safety-adjacent semantics.
> >
> > **(2) The two-stage filtering process prevents misclassification of multi-use features.**
> > Features with mixed semantics may appear in the high-|s_i| candidate pool, but the second-stage verification—based on activation-context inspection—allows us to detect when a feature appears across both benign and harmful contexts. Such features are consistently evaluated as having ambiguous relevance and are filtered out. In our updated appendix, we include human–GPT-4o side-by-side annotations showing that mixed-use features are given intermediate scores, not “forced” into safe or unsafe categories.
> >
> > **(3) SAFER targets dataset-level safety signals, not universal semantic labels.**
> > Our goal is not to assert that a feature is intrinsically safe or unsafe across all contexts, but to quantify how that feature contributes to safety distinctions within the reward model’s training distribution. If a feature occurs in diverse benign settings, then its activations won’t dominate the contrastive signal, and SAFER correspondingly treats it as low-impact for safety alignment—precisely the desired behavior when modifying preference datasets.
> >
> > To further address the reviewer’s concern, **we have clarified in the revision that SAFER is designed to identify safety-discriminative features, not to perform universal semantic labeling.** Features whose semantics depend heavily on context will naturally yield small contrastive scores and are therefore not used for interventions such as denoising. This avoids the pitfall of penalizing the model’s ability to handle sensitive topics in benign contexts. We agree that more fine-grained, context-sensitive feature analysis is an important direction and explicitly note this as future work.

---

### Official Review · Reviewer_u1t1 · 2025-11-03

**Soundness:** 2
**Presentation:** 3
**Contribution:** 2
**Rating:** 6
**Confidence:** 2

**Summary:**

The paper introduces SAFER (Sparse Autoencoder For Enhanced Reward model) , a framework designed to address the "black box" nature of reward models used in RLHF. The authors propose that by applying sparse autoencoders to the internal activations of a safety-trained RM, they can extract sparse, human-interpretable, monosemantic features that correspond directly to safety-related concepts.

The method involves training an SAE on RM activations and then using a contrastive method to identify features that have the largest activation difference between "chosen" (safe) and "rejected" (unsafe) responses. These features are filtered using GPT-4o to isolate a core set of safety-relevant features. This approach is validated through two primary experiments: (1) flipping the labels of the top-scoring safety-aligned data points can degrade the RM's safety performance, (2) conversely, removing the lowest-scoring (noisiest) data points can improve the RM's safety performance.

**Strengths:**

1. The proposed method (SAFER) is a precise and targeted intervention. Figure 1 and Table 2 show that SAFER's poisoning method causes a sharp decline in safety scores while leaving chat performance almost untouched.
2. SAFER can also work as a filtering method. In the denoising experiments, it shows an improvement in RM safety from 94.86 to 96.46 on the 3B model by removing just 4% of data.
3. Ablation experiments are sufficient; having a separate discussion on token-level and sequence-level is a plus.

**Weaknesses:**

1. A significant methodological weakness is the reliance on a proprietary black-box GPT-4o to filter the features. The framework first uses a contrastive score to find candidate features, but then "use GPT-4o to interpret and assign safety relevance ratings" and only retains features with a perfect score of 5. Though the authors validate Human-GPT-4o alignment (Figure 5), but this is on their specific task, and it doesn't address the dependency.
2. The effect of removing denoising data does not monotonically increase as more "noisy" data is removed. For both the 1B and 3B models, the peak safety performance occurs at a 4% removal rate, after which performance slightly degrades or fluctuates. This suggests that the $score_{safe}$ metric is not a perfect proxy for "data quality" and that simply removing all low-scoring samples may not be the optimal strategy. The paper does not fully explore or explain this non-monotonic behavior.
3. Lack of scale. Features that are clearly separable in small models may become more polysemantic or distributed via superposition in larger models. Will this method of extracting "monosemantic" safety features work on larger models? (e.g. Qwen-3-8B, Llama-3.1-70B)

**Questions:**

See weakness.

---

> ### Author Response · Authors · 2025-11-22
> **Response to Reviewer u1t1 (Part 1/2)**
>
> > Q1: A significant methodological weakness is the reliance on a proprietary black-box GPT-4o to filter the features. The framework first uses a contrastive score to find candidate features, but then "use GPT-4o to interpret and assign safety relevance ratings" and only retains features with a perfect score of 5. Though the authors validate Human-GPT-4o alignment (Figure 5), but this is on their specific task, and it doesn't address the dependency.
>
> We thank the reviewer for highlighting the concern regarding the use of GPT-4o in the feature-filtering stage. We agree that, in principle, human evaluation would be the most reliable method for assessing the safety relevance of extracted features. However, performing detailed annotation for thousands of SAE features is prohibitively time-consuming and is generally considered infeasible in current mechanistic-interpretability workflows. Consequently, recent studies on large-scale SAE interpretation similarly rely on LLM-as-judge pipelines for feature validation [1, 2].
>
> To mitigate this dependency, our framework adopts a **two-stage filtering process**:
>
> 1. **Contrastive extraction (model-only):**
>  The primary identification of safety-relevant candidates is entirely **model-driven**, using contrastive activation differences between chosen and rejected responses. This step already isolates features that are strongly tied to RM's internal safety signals.
> 2. **LLM-based verification only for top-ranked features:**
>  GPT-4o is used solely to **validate** the semantic meaning of the top-|sᵢ| features by inspecting their activation contexts, ensuring they correspond to genuine safety-related concepts rather than spurious correlations.
>
> To evaluate whether this verification step introduces systematic bias, we conducted human–GPT-4o comparison on randomly sampled features (Figure 5). **As added in the updated appendix, we also include side-by-side human explanations for several representative features.** Across all cases, human and GPT-4o assessments align closely, and we did not observe meaningful conceptual discrepancies. This suggests that GPT-4o serves as a practical proxy for large-scale semantic annotation rather than a critical dependency that shapes the overall results.
>
> Importantly, the conclusions of SAFER do not rely on GPT-4o discovering new features: it merely helps **confirm** whether contrastively extracted features genuinely relate to safety. The core methodological insight—that safety-relevant features can be identified via activation-level contrast—holds independently of the LLM-based verification step.

---

> > ### Author Response · Authors · 2025-11-22
> > **Response to Reviewer u1t1 (Part 2/2)**
> >
> > > Q2: The effect of removing denoising data does not monotonically increase as more "noisy" data is removed. For both the 1B and 3B models, the peak safety performance occurs at a 4% removal rate, after which performance slightly degrades or fluctuates. This suggests that the score_safe metric is not a perfect proxy for "data quality" and that simply removing all low-scoring samples may not be the optimal strategy. The paper does not fully explore or explain this non-monotonic behavior.
> >
> > We thank the reviewer for the insightful observation. We agree that safety performance peaks around a 4% removal rate and then exhibits mild fluctuations. This non-monotonic behavior is expected in selective denoising. **Removing a small fraction of genuinely noisy or misaligned pairs improves safety alignment, but more aggressive removal inevitably discards informative but rare boundary cases that contribute to the reward model’s generalization ability.** Thus, performance does not increase indefinitely with more data removed.
> >
> > Importantly, the same trend is also observed in the Reward-based baseline (Table 3), suggesting that this pattern stems from the intrinsic structure of preference datasets rather than a deficiency in the proposed score_safe metric. Our metric is specifically designed to rank samples by safety relevance—not to serve as a monotonic estimator of global data quality. **We have clarified this interpretation in the revised manuscript.**
> >
> > > Q3: Lack of scale. Features that are clearly separable in small models may become more polysemantic or distributed via superposition in larger models. Will this method of extracting "monosemantic" safety features work on larger models? (e.g. Qwen-3-8B, Llama-3.1-70B)
> >
> > We appreciate the reviewer’s concern regarding scalability. **Large-scale validation is indeed computationally demanding, especially since each poisoning or denoising ratio requires retraining a separate reward model.** This constraint limited our experiments to 1B and 3B models.
> >
> > That said, prior work on sparse autoencoders [1, 2] provides strong evidence that monosemantic features reliably emerge—and often become more distinct—as model size increases, despite greater superposition. SAFER directly inherits this scalability: it operates on hidden activations rather than model parameters, making the method architecture-agnostic and naturally compatible with larger reward models (e.g., Qwen, Llama families).
> >
> > While full-scale experiments are left as future work, the underlying SAE mechanisms supporting SAFER have already been shown to extend effectively to models far larger than those we evaluate.
> >
> > [1] Anthropic, 2024; Scaling Monosemanticity: Extracting Interpretable Features from Claude 3 Sonnet
> >
> > [2] Cunningham et al. 2023; Sparse Autoencoders Find Highly Interpretable Features in Language Models

---

### Author Response · Authors · 2025-12-01
**General Response**

We sincerely thank all reviewers for their thorough, constructive comments and valuable suggestions. We are encouraged by the positive feedback on our work’s core strengths, including rigorous experimental design, clear presentation, and innovative targeted reward model safety manipulation via Sparse Autoencoders. Notably, **Reviewer Wp5f has raised the score following our revisions**, which we greatly appreciate.

To facilitate clarity, we note the abbreviations used in the summary table: **R = Reviewer** (followed by ID and comment ID), **S = Strength** (positive reviewer feedback),  and **W = Weakness/Concern** (points for clarification or supplementation). Below, we summarize key strengths, concerns, our clarifications, and supplementary experiments added in the revision.

| | **Presentation** | **Novelty** | **Experimental Insights** | **Experimental Details** | **Evaluation** |
|---------------------------------------------|--------------------------------------------------------------------------|---------------------------------------------------------------------------|----------------------------------------------------------------------------------------------|-----------------------------------------------------------------------------------------|---------------------------------------------------------------------------------------------------------------------------|
| **Reviewer#1 u1t1** | **W2:** Requests explanation for non-monotonic denoising effects. | NA | **W1:** Concern about reliance on proprietary GPT-4o for feature filtering. | NA | **W3:** Lack of scalability to larger models. |
| **Reviewer#2 fdCC** | NA | **S1:** Values application of mechanistic interpretability to RMs. | **S3:** The experimental design and validation are rigorous and standardized. | **W2:** Requests causal intervention experiments to prove feature causality. | **W3:** Requests clarification on feature selection strategy. |
| **Reviewer#3 88cy** | **S4:** Readable and reproducible. | **W1:** Limited novelty, mainly applying SAE to reward models. | **W4:** Requests generality to other alignment aspects (helpfulness/honesty). | **W5:** Requests more visualizations. | NA |
| **Reviewer#4 Wp5f (score ↑)** | **S3:** Text flow is logical and figures are clearly described. | **S1:** Novel method for targeted RM safety manipulation via SAE. | **W1:** Justify SAE over simpler alternatives. | **W2:** Requests results on Chat Hard and Math subsets. | **W3:** Explicitly link reward-based baseline to prior work. |
| **[Action] Clarification** | **R1W2:** Explained non-monotonic denoising as intrinsic dataset structure. | **R3W1:** Articulated novelty as "mechanistic activation contrast + targeted data manipulation". | **R1W1:** Clarified GPT-4o’s role (two-stage filtering + human alignment validation). | NA | **R1W3:** Clarified scalability via SAE’s architecture-agnostic design; **R2W3:** Clarified feature selection; **R4W3:** Linked reward-based baseline to Pathmanathan et al. (2024). |
| **[Action] Supplementary Experiments** | NA | NA | **R3W4:** Extended to helpfulness RMs via HelpSteer2 dataset; **R4W1:** Added SAFER vs. GPT-4o-mini/Detoxify comparative experiments. | **R2W2:** Conducted feature-steering causal experiments; **R3W5:** Added UMAP cross-model safety-feature clustering visualizations; **R4W2:** Supplemented Chat Hard/Math subset scores for poisoning/denoising. | NA |


We hope that our detailed responses and targeted revisions have adequately addressed all reviewers’ concerns. We again thank the reviewers for their insightful feedback, which has significantly improved the quality of our work, and we welcome further discussion if needed.

-- Best,

Authors of Paper# 15398

---

### Meta-Review · Area_Chair_hRBj · 2025-12-17

**Summary:**

This paper introduces SAFER (Sparse Autoencoder For Enhanced Reward model), a framework for interpreting and improving reward models through mechanistic analysis using Sparse Autoencoders (SAEs) to uncover human-interpretable safety-relevant features. The method applies contrastive feature extraction to identify features with largest activation differences between preferred and dispreferred responses, then uses these for targeted data poisoning and denoising strategies. The paper received initial ratings of 4, 4, 6, and 6 from four reviewers, with Reviewer Wp5f raising their score from 4 to 6 after rebuttal.

The primary strengths include rigorous experimental design with comprehensive ablations on SAE hyperparameters (layer positions, dictionary size, sparsity), innovative targeted manipulation demonstrating that SAFER can precisely degrade or enhance safety alignment with minimal data modification without sacrificing general performance. Authors also provided extensive rebuttals with substantial supplementary experiments.

However, significant concerns remain about limited novelty and scope. Reviewers noted the work mainly applies existing SAE methods to reward models with modest core algorithmic contribution—the novelty lies primarily in the problem setting (applying SAEs to reward models rather than base LLMs) and contrastive feature extraction pipeline rather than new SAE architecture. Reliance on GPT-4o for feature filtering raised methodological concerns. I agree with reviewers that the main technical contribution is limited. For example, would applying other interpretability methods (e.g., PCA, ICA) yield similar insights? The paper would benefit from clearer articulation of how SAFER advances beyond standard SAE applications and more thorough comparison to alternative interpretability techniques. I recommend rejection.

**Reviewer Concerns:**

Partially addressed.

**Reviewer Scores:**

One reviewer agreed to raise the score from 4 to 6.

---

### Decision · Program_Chairs · 2026-01-26

Reject